# Analysis of Sequence Divergence in Mammalian ABCGs Predicts a Structural Network of Residues That Underlies Functional Divergence

**DOI:** 10.3390/ijms22063012

**Published:** 2021-03-16

**Authors:** James I. Mitchell-White, Thomas Stockner, Nicholas Holliday, Stephen J. Briddon, Ian D. Kerr

**Affiliations:** 1School of Life Sciences, University of Nottingham, Queen’s Medical Centre, Nottingham NG7 2UH, UK; nicholas.holliday@nottingham.ac.uk (N.H.); stephen.briddon@nottingham.ac.uk (S.J.B.); 2Centre of Membrane Proteins and Receptors, Universities of Birmingham and Nottingham, The Midlands, Nottingham NG7 2UH, UK; 3Center for Physiology and Pharmacology, Institute of Pharmacology, Medical University of Vienna, Währingerstrasse 13A, 1090 Vienna, Austria; thomas.stockner@meduniwien.ac.at

**Keywords:** ABC transporter, multidrug resistance, membrane protein, functional divergence, ABCG2

## Abstract

The five members of the mammalian G subfamily of ATP-binding cassette transporters differ greatly in their substrate specificity. Four members of the subfamily are important in lipid transport and the wide substrate specificity of one of the members, ABCG2, is of significance due to its role in multidrug resistance. To explore the origin of substrate selectivity in members 1, 2, 4, 5 and 8 of this subfamily, we have analysed the differences in conservation between members in a multiple sequence alignment of ABCG sequences from mammals. Mapping sets of residues with similar patterns of conservation onto the resolved 3D structure of ABCG2 reveals possible explanations for differences in function, via a connected network of residues from the cytoplasmic to transmembrane domains. In ABCG2, this network of residues may confer extra conformational flexibility, enabling it to transport a wider array of substrates.

## 1. Introduction

ATP-binding cassette (ABC) proteins form a very large family across all domains of life, responsible for the primary active uptake and export of nutrients, toxins, lipids, peptides and other metabolites. ATP hydrolysis is carried out at two cytoplasmic nucleotide-binding domains (NBDs), and the energy released is coupled to conformational changes in two transmembrane domains (TMDs) to power transport of substrate. In mammals, ABC proteins are divided into seven subfamilies, A–G, although the ABCE and F families lack TMDs and are associated with ribosome function [1]. Often, members within a subfamily, though sharing common descent, can have very different functions. The family investigated here is the G subfamily of ABC transporters in mammals (ABCGs). In most mammals, there are five members of this subfamily [2]. All mammalian ABCGs share a common arrangement of domains, all being “half-transporters” with just a single NBD and TMD in the primary amino acid sequence. A unique property of ABCG arrangement is that the NBD is N-terminal to the TMD, so they are referred to as “reverse” half-transporters.

Four of the mammalian ABCGs have a repertoire of substrates limited to lipids. Two of these, ABCG1 and ABCG4, have sequences much more similar to one another than they are to the rest of the ABCGs. They also seem to share much of their function, regulating cholesterol metabolism by transporting cholesterol into high-density lipoprotein [3]. Precise differences in their function are yet to be determined, but they do seem to differ significantly in their tissue expression profiles [3,4,5].

The two other lipid-transporting ABCGs, ABCG5 and ABCG8, are also more closely related to one another than they are to the other ABCGs, though to a lesser extent than ABCG1 and ABCG4. They have taken on necessarily different roles by forming an obligate heterodimer, neither protomer being trafficked to the membrane if expressed alone [6]. ABCG5/G8 expressed in the liver and intestine limits the uptake of toxic plant and shellfish sterols and is responsible for 35% of the efflux of cholesterol in the intestine. The ABC dimer G5/G8 has only one functional ATP-binding site, indicating that ABCG5 and ABCG8 have diverged in function in this respect.

The final ABCG, ABCG2, has a much broader substrate specificity. It was first isolated in placental tissue and breast cancer cell lines [7,8], and has been since identified as a multidrug resistance (MDR) protein. It can export a wide variety of substrates, including many chemotherapy drugs, making it a target of great therapeutic interest. For this reason, it is the best studied of the ABCG subfamily.

Recently, structures have been solved for ABCG5/G8 and ABCG2. First came the structure of ABCG5/G8 [9], which was used to model a structure of ABCG2 [10]. Docking substrates to this model identified multiple possible binding sites, already suggested by previous biochemical work [11]. With the first structures of ABCG2 [12,13,14], its broader substrate specificity was explained through a relatively large internal cavity, compared to ABCG5/G8′s deep, slit-like cavity, forming part of the transport pathway, though in more recent structures, the cavity is only present in structures with substrates bound [15]. In spite of these structural advances, the molecular basis for differences in function between ABCG family members is largely unknown. As their differences ultimately arise from differences in their sequence, it is possible that comparison of conservation between ABCGs could provide clues to help ascertain this molecular basis.

Families of genes can occur when a gene duplicates and the different copies start to take on different functions, a process known as functional divergence [16,17]. When this happens, the evolutionary pressures on the duplicated genes start to differ, with impact on the sequences of the proteins encoded. Non-synonymous mutations in structural elements with functional importance are less likely to persist [18]. In two functionally divergent proteins, a structural element may be more important to the function of one than the other, which will be reflected in this region being better conserved in the protein for which the element is more important. This has been called type I divergence [16]. A similar phenomenon, type II divergence, occurs if the same element is important for the function of both proteins, but the important properties of the amino acid found there are different. This is reflected in the region being conserved in both proteins, but with different amino acids being conserved. The differences in sequence conservation caused by functional divergence have been used to identify important sites in proteins.

In order to analyse the conservation between the members of the G subfamily of ABC transporters in mammals, we have calculated functional divergence of residues based on Shannon entropy [19] from a large multiple sequence alignment of ABCGs. We have examined residues with particular patterns of type II divergence between ABCGs reflecting some of the functional divergence responsible for their differences. Hypotheses regarding the structural basis of these functional differences were derived by mapping positions in the alignment that share particular types of conservation onto the apo-closed structure of ABCG2. Specifically, we have identified a top-to-bottom signature, passing through the polar relay of ABCG5/G8 [9,20], which may contribute to allosteric differences in the G subfamily responsible for differences in substrate specificity.

## 2. Results

### 2.1. Overall Conservation Patterns

A total of 174 ABCG protein sequences (summarised in Appendix A) were analysed. These were grouped according to the protein they represent, and their conservation calculated as described in methods. A tree constructed from these sequences showing the relationship between the ABCG proteins is shown in Figure 1a. The alignment had a length of 1269 positions (henceforth “columns”). Of these, 674 columns had gaps in either >10% of all sequences or >30% of sequences for one of the proteins (see Appendix A). Of the remaining 595 columns, 594 met the entropy cutoff for conservation in at least one protein. A total of 61 of these columns were conserved across the ABCG family, and the remaining 533 had some type of divergence, as summarised in Figure 1b.

An example of conservation is represented in Figure 2. It shows one part of the interface between TMD and NBD which is vital for transmitting energy from ATP hydrolysis in ABCGs, often referred to as the “elbow helix” in ABCG literature. Columns in this region of the alignment display the different types of conservation of relevance; firstly columns that show total conservation, where not only is the column conserved for each protein, but it is conserved in the same way (e.g., column 900 in Figure 2 where all ABCG sequences conserve arginine at this position). Secondly, it shows type I divergence, where the column is conserved as the same amino acid for at least one protein, with other proteins not conserving the column. For example, column 895 in Figure 2 is conserved as a cysteine in ABCG1 and ABCG4 but is not conserved in ABCG2, ABCG5 or ABCG8. Thirdly, type II divergence, where each protein shows conservation, but different proteins can be conserved in different ways is evident in columns 893, 894, 897, 901, 904 and 905 in Figure 2. For example, in position 905, ABCG1 and ABCG4 conserve isoleucine, ABCG2 and ABCG5 conserve leucine and ABCG8 conserves aspartate. Finally, several other columns display a mix of types of divergence; for example, column 890 shows a position conserved only in ABCG1, ABCG2 and ABCG4, and the residue conserved is different in all cases.

Many of the approaches used to investigate functional divergence return a score for each column reflecting how it is conserved across the whole alignment and some are limited to a comparison between two groups. In the case of the ABCGs, one aspect of functional divergence worth exploring might be ABCG2′s broader substrate specificity. If comparing two groups, examining both type I and type II divergence is worthwhile. However, the substrates transported by ABCG1 and ABCG4 differ from those transported by ABCG5/G8, so their substrate specificities are achieved in different ways. Considering possible functional divergence within ABCG members highlights some of the difficulties with terminology. Here, we have defined type II divergence to include any column in which each protein is conserved, without conservation across the whole family, and type I divergence to include columns in which one or more proteins have the same amino acid conserved, and all other proteins are not conserved. Rather than calculating scores for the whole alignment, we have classified columns according to the proteins in which they are conserved, allowing inferences to be drawn from differences between multiple groupings.

In this manuscript, the different ways to group proteins to examine their conservation is referred to as a conservation pattern. Columns with a particular conservation pattern are represented by having any family members conserved in that column written in brackets. If more than one member has the same amino acid conserved at that position, they are held in the same brackets, separated by a comma. To illustrate this nomenclature with respect to Figure 2, among the conservation patterns visible in this section of the alignment are (ABCG1, ABCG4), (ABCG2), (ABCG5), (ABCG8) in column 901 and (ABCG1, ABCG4) (ABCG2, ABCG5, ABCG8) in column 904, both of which represent type II divergence.

There are 202 theoretically possible conservation patterns, of which around half (106) are observed anywhere in the actual alignment. Most of these have very few representatives, with over 60 only having 1–3 representatives. Remarkably, almost half of the divergent positions in the alignment are contributed by just 14 different conservation patterns (Appendix A). Some of these well-represented patterns have implications for functional divergence when the relationships between the proteins are considered.

### 2.2. Phylogenetically Significant Type II Divergence in ABCGs

To explore the differences in substrate specificity within the G subfamily, it is necessary to explore columns of the alignment showing functional divergence. Residues essential to maintaining the overall ABCG fold will be either identical across proteins or highly conserved. Other behaviours, such as force transmission and substrate recognition, are likely to be conserved by each protein, but change across the family. The approach adopted here, which classifies columns by their conservation pattern, was deliberately chosen to allow interpretation of differences between multiple groups within the alignment. Though it does not provide a score, classifying columns by conservation pattern allows discrimination of functional divergence at different levels, exploiting existing knowledge of the proteins under investigation. Emphasis here was on the ability to estimate functional divergence in a way that allows interpretation based on what we know of the proteins involved.

The conservation patterns that are most likely to yield insight into differences in substrate binding are those that separate proteins by their substrates. ABCG1 and ABCG4 have a high sequence identity, and are identical in 434 of 595 columns (excluding gapped columns). ABCG1 and ABCG4 also overlap in their function and substrates [3,21], so grouping them together to establish functional divergence is sensible from both an evolutionary and functional perspective. Though ABCG5 and ABCG8 by definition transport the same substrates, their interactions with those substrates may differ, if the shape and chirality of the substrate is reflected in the substrate-binding site. Furthermore, they are less similar in sequence (being conserved in the same way in only 138 columns), and to some extent must carry out different functions due to the asymmetry of their nucleotide-binding sites [6].

### 2.3. The Conservation Pattern (ABCG1, ABCG4), (ABCG2), (ABCG5), (ABCG8) Defines a Possible Allosteric Pathway in ABCG Proteins

The pattern (ABCG1, ABCG4), (ABCG2), (ABCG5), (ABCG8) separates the proteins by their probable substrate interactions, and is the most populated set of functionally divergent columns, with 33 columns. Residues corresponding to these columns are mapped on to an ABCG2 structure in Figure 3a. In this structure of ABCG2 (and also observed if mapped onto ABCG5/G8, Appendix A), these residues form a “corkscrew” pattern from the cytoplasmic face of the NBDs, through the TMDs to the extracellular face of the protein. This distribution implies that some important differences in the function of members the G subfamily are due to differences in allostery, as corresponding residues are ideally placed to form a network of residues coordinating conformational changes throughout the protein. Their distribution can be compared with residues with the conservation pattern (ABCG1, ABCG4), (ABCG2), (ABCG5, ABCG8) (Figure 3b) and the much less common pattern (ABCG1), (ABCG4), (ABCG2), (ABCG5), (ABCG8) (Figure 3c).

### 2.4. Conservation of the Polar Relay

A feature of ABCG5/G8 identified from the ABCG family fold [9] that is also likely to carry out a role in conformational changes is the “polar relay”. This comprises 11 residues from ABCG5 and 9 residues from ABCG8. In the multiple sequence alignment, five of these positions overlap, leaving a total of 15 columns in the alignment corresponding to the polar relay (Figure 4). Notably, one of the columns in the polar relay of both ABCG5 and ABCG8 (column 1011) aligns with R482 in transmembrane helix 3 of ABCG2, mutation of which has long been shown to alter substrate specificity [11,22,23].

The conservation patterns of these fifteen residues are shown in Figure 4. Though 12 of these positions have type II divergence for all ABCGs, two columns (915: R389 in ABCG5 and H420 in ABCG8; and 963: N437 in ABCG5 and D466 in ABCG8) are not conserved in ABCG8, and one is not conserved in ABCG5 (1006: V471 in ABCG5 and E500 in ABCG8). One remarkable observation is that 40% of the columns in the polar relay (6/15) have the conservation pattern (ABCG1, ABCG4), (ABCG2), (ABCG5), (ABCG8). This makes this conservation pattern much more common here than in the whole protein, as it is only found in 5.5% (33/595) of the aligned columns (Appendix A).

The observation that the type II divergence pattern (ABCG1, ABCG4), (ABCG2), (ABCG5), (ABCG8) subsumes much of the polar relay (as shown in Figure 4 and Figure 5a), which has previously been attributed allosteric significance in ABCG5/G8, suggests that the entire corkscrew of residues contributes to the allosteric divergence of the ABCG family.

### 2.5. Sidechain Properties in the Allosteric Corkscrew

In a recent review [24], it was noted that composition of residues in the polar relay of ABCG2 and of ABCG5/G8 differs, with ABCG8 having a relatively high number of charged residues and ABCG2 a relatively low number. In residues with the conservation pattern (ABCG1, ABCG4), (ABCG2), (ABCG5), (ABCG8) this pattern is reiterated, ABCG8 having seven charged residues, ABCG5 five and ABCG2 one (Figure 5b). Perhaps notably, the only charged residue in ABCG2 with this conservation pattern is R482, mutation of which is long associated with altered substrate specificity [11,22,23].

An even greater discriminant between ABCG2 and other ABCG members is that 15 of the 33 residues with this conservation pattern are polar, hydroxylated residues (serine or threonine) in ABCG2 (Appendix A). There are relatively few hydroxylated amino acids in these positions for other ABCGs (ABCG1/4: 5, ABCG5: 5, ABCG8: 3). Do these dissimilarities in the corkscrew of type II divergent residues contribute to differences in protein function? Polar and ionisable residues can drive specific helix oligomerisation, but this does not include serine or threonine alone [25,26,27] and we did not observe the specific motifs predicted to drive helix association [28,29]. Rather, an intriguing possibility is that serine and threonine form intra-helical hydrogen bonds, which can bend the helix in certain conformations, lending ABCG2 unusual flexibility in this region. Other residues significantly contributing to flexibility, such as glycine and proline, are no more common in other proteins compared with ABCG2. This extra flexibility could permit the binding and transport of diverse sizes of substrates, coupled to allosteric motions communicated through this network. Similar influence of hydroxylated amino acids in driving substrate-specific conformational changes is observed in some GPCRs [30,31,32,33].

### 2.6. Conservation of Other Regions

Electron microscopy and X-ray crystallography data on ABCGs have indicated other structural regions that are proposed to be critical for allosteric communication. We analysed the triple helical bundle between the NBD and TMD, which is considered to be a vital region for transmission of force from ATPase activity to the TMD. This region spans 54 columns in the multiple sequence alignment and 28 different conservation patterns are observed here. The hot-spot helix is most highly conserved, with 40% of its residues being conserved across the alignment, but no other patterns are significantly different here from the alignment as a whole (Appendix A). Though the triple helical bundle is highly conserved, the whole of it is not conserved across the G subfamily. Nor is it a motif that defines the difference between ABCG members well. Part of this comes from its being less well conserved in ABCG5 and ABCG8 (~70% for each), perhaps indicating that heterodimerisation reduces the evolutionary pressure on some of these positions. This may be particularly true for this region, due to its importance for transmitting force from ATP hydrolysis [34], which is altered in ABCG5/G8 due to the degenerate NBS.

Given the differences between the dimerisation behaviour of ABCG members (i.e., that some form homodimers and others are obligate heterodimers), we inspected the dimerisation interfaces (both at the TMD:TMD and the G-family specific NBD:NBD interface [35] to see if this was reflected in the conservation patterns. Residues within 5 Å of the other protomer in the structures 6VXF (ABCG2) and 5DO7 (ABCG5/G8) were found and the conservation patterns of corresponding columns were examined (Appendix A). A total of 46 different patterns are represented in this set, with completely conserved again being the most frequently observed. However, none of the patterns makes up a statistically significant fraction, meaning that the dimer interface is not a useful discriminant between ABCG members.

Binding pockets for substrates of ABCG2 and ABCG5/G8 have been identified from their structures. These are compared in Appendix A. Interestingly, there is little overlap between the residues contributing to these pockets, with two columns contributing to the binding pockets of both ABCG2 and ABCG5, and two contributing to both ABCG5 and ABCG8. Both of the columns contributing to the pockets of both ABCG2 and ABCG5 have the conservation pattern (ABCG1, ABCG4), (ABCG2), (ABCG5), (ABCG8), so are part of the corkscrew. A further five columns with this conservation pattern contribute to the binding pocket of ABCG2, and another to that of ABCG5.

### 2.7. Other Conservation Patterns

Though the conservation pattern (ABCG1, ABCG4), (ABCG2), (ABCG5), (ABCG8) is the most observed in functionally divergent columns, some other patterns are well represented (the frequencies of well-represented functionally divergent conservation patterns are shown in Appendix A, and some of these are represented on the structure of ABCG2 in Appendix A). Given the evolution of the subfamily, it is instructive to examine the conservation pattern (ABCG1, ABCG4), (ABCG2), (ABCG5, ABCG8), which highlights another 13 residues, shown on ABCG2 in Figure 3b. Notably, ten of these are found either in the NBD:NBD interface or the NBD:TMD interface. The remainder are found in the TMD, and two of these (C438 and I573 in ABCG2, P431/460 and F567/595 in ABCG5/G8) form pairs in the structures of ABCG2 and ABCG5/G8.

An interpretation of these patterns based on their likely evolution is that both of these sets of residues diverged when the ancestors of ABCG1 and ABCG4, ABCG2, and ABCG5 and ABCG8 specialised to transport different substrates. Later, ABCG5 and ABCG8 could take on different parts of the function of a transporter by forming an obligate heterodimer, and the residues corresponding to columns conserved as (ABCG1, ABCG4), (ABCG2), (ABCG5), (ABCG8) represent further functional divergence. Thus, both sets would be responsible for differences in substrate specificity, with 13 residues requiring conservation across ABCG5 and ABCG8. Taken together, these patterns reiterate the likely importance of allostery to differences in the function of the ABCGs.

The three patterns found most frequently other than (ABCG1, ABCG4), (ABCG2), (ABCG5, ABCG8), having 24, 23, and 23 members respectively, are (ABCG1, ABCG4), (ABCG2), (ABCG5); (ABCG1, ABCG4), (ABCG2), (ABCG8); and (ABCG1, ABCG4), (ABCG2). In total, these four patterns make up 103/533 of the functionally divergent columns. In all of these, ABCG1 and ABCG4 conserve the same amino acid, and ABCG2 conserves another, but conservation within ABCG5 and ABCG8 differs. In the conservation patterns not examined more closely in the sections above, ABCG5 or ABCG8 or both do not conserve the column. This indicates positions which have a decreased evolutionary pressure in ABCG5 and ABCG8, perhaps due to their splitting some of the functions which normally both halves of a dimer must maintain due to their forming a heterodimer. That so many of these positions are also sources of functional divergence between ABCG1 and ABCG4 on one hand and ABCG2 on the other is intriguing.

Another set of conservation patterns that is well represented is columns with type II divergence between one member and all the other members. (ABCG1, ABCG4, ABCG2, ABCG8), (ABCG5) has 16 members. Two interesting residues with this pattern are: F439 in ABCG2 (a tyrosine in ABCG5), which serves as a “clamp” for substrates [36], and E451 in ABCG2 (a leucine in ABCG5), which is a key residue in coupling ATPase activity to transport [34]. (ABCG1, ABCG4, ABCG5, ABCG8), (ABCG2) has 12 members. (ABCG1, ABCG4, ABCG2, ABCG5), (ABCG8) has 9 members. Due, probably, to the close relatedness of ABCG1 and ABCG4, there are fewer (0 and 5 respectively) columns with type II divergence between these and the rest of the subfamily. These are tantalising groups, as they show places that each member specialises in a way distinct from the ABCG family on the whole. However, a molecular interpretation is much more difficult.

## 3. Discussion

In this study, we have identified a corkscrew region in ABCG transporters whose conservation suggests a role in substrate specificity. Though no experimental work has yet been carried out to deliberately explore the functional effects of mutations to the corkscrew, some of its residues have been mutated as part of other studies, or observed as naturally occurring single nucleotide polymorphisms (Appendix A). Particularly notable in this regard is Cox et al. 2018 [37], which includes mutagenesis of five residues with the conservation pattern (ABCG1, ABCG4), (ABCG2), (ABCG5), (ABCG8), including the very well-studied residue R482A. Mutations of three of these five (T402A, S440A, and I543A) compromise transport of both mitoxantrone and pheophorbide A. T402 mutations have previously been described [38,39] as having decreased transport activity. Several others have observed diminished transport by ABCG2 with mutation of these residues, including mutations to S384, T434, and S441 [38,40,41,42,43,44,45]. Recently, diminished ATPase activity has been observed in ABCG5 [20] with mutation of A540 to phenylalanine, a residue also sharing this conservation pattern.

Other mutagenesis studies have included residues identified in this analysis as conserved in all aligned proteins. Many of these result in poor expression of mature protein [34,37,46], such as mutation of E138 in ABCG2 [34]. Though some have discernible effects on transport, surprisingly, mutation of P480 to alanine, despite being a mutation to a residue conserved in all sequences used in this analysis, and with dramatic chemical differences, has no effect on transport in ABCG2 [37,47]. This provides a cautionary example that care must be taken when interpreting these results. Other mutations to these positions are found as variants in vivo, some causing sitosterolemia, such as mutations to E146 in ABCG5, analogous to E138 in ABCG2 [48,49,50,51,52,53,54,55,56,57]. A summary of this, including disease-causing variants, can be found in the Appendix A.

Comparing the ways mammalian ABCGs are conserved shows functionally and evolutionarily important signatures that are well represented in previous mutagenesis studies. Differences in the substrate specificity of subfamily members correspond to patterns of conservation that, when mapped onto 3D structures, are ideally placed to modulate the communication of conformational change between domains, suggesting that this may be responsible for some of the differences in substrate specificity. Particularly, grouping ABCG1 and ABCG4 together identifies a pattern, which we have named the corkscrew network. This is also important to a previously identified structural feature, the polar relay, and suggests a unifying hypothesis for substrate specificity in the subfamily: that allostery in this network underlies functional divergence. Appropriate experiments to test the importance of the corkscrew network of residues to differences in ABCG function promise to reveal interesting factors in their transport mechanism.

## 4. Materials and Methods

### 4.1. Sequence Acquisition

Through the NCBI, the RefSeq database [58] was queried for all nucleotide sequences matching “ABCG AND mammalia [organism]”. Analysis was restricted to mammalia to afford greater confidence that function corresponded to the identity of the protein. Initially, 778 sequences from 112 species were identified. Not every species had a full complement of sequences for ABCG1, ABCG4, ABCG2, ABCG5 and ABCG8, so, where possible, these were found in the RefSeq database and added manually. A matching list of protein sequence IDs were used for a submission to Entrez. An in-house Python [59] script was used to check for and remove identical sequences.

Further sequences from some species were removed to prevent sequences from closely related species biasing later analysis. For example, sequences from 29 primates made up a high proportion of the total number of sequences, but presumably a low proportion of the organismal diversity. For this reason, 25 of the sequences were removed, keeping one ape (*Homo sapiens*), one monkey (*Piliocolobus tephrosceles*), one gelada (*Theropithecus gelada*), and one lemur (*Microcebus murinus*). Similar reasoning was used to reduce the number of species to 40. When choosing species to keep, a series of criteria were used. First, any well-studied species (e.g., *Homo sapiens, Mus musculus*) were retained. Next, species where one or more ABCG sequences were only tentatively identified (e.g., deposited in the database with the caveat “LOW QUALITY PROTEIN”, or that were somewhat shorter than the canonical length of ABCGs (ca. 650 amino acids) were eliminated in preference to species with higher quality sequences. A preliminary alignment of all sequences using multiple alignment fast Fourier transform (MAFFT) was performed. This alignment was processed using MaxAlign, which identifies sequences that align most poorly with the others. If sequences from a species aligned poorly, they were disfavoured in the elimination process. In some cases, a species without an obvious substitute was eliminated—for example, the African elephant has only two ABCG sequences and both are low-quality sequences which aligned poorly. For this reason, the final number of species was reduced to 35. Where species could not be distinguished using these criteria, a random integer between one and the size of the set being reduced was generated, and the sequence matching that number in alphabetical order was kept. A summary of the sequences used can be found in Appendix A.

### 4.2. Alignment and Tree Construction

The final 174 protein sequences were aligned with MAFFT using the automatically assigned strategy, and other parameters set automatically by the MAFFT server, except raising the offset value to 0.123, which is the default value for the command line tool. This alignment was used to construct a tree using the Simply Phylogeny tool from ClustalW2, which was then visualised with the interactive Tree of Life [60]. The large number of sequence names reduced the clarity of the figure, so were removed, but the branches were otherwise left intact.

### 4.3. Calculation of Conservation

First, columns in which at least 10% of the total alignment, or 30% of one protein (e.g., ABCG1 sequences) were gaps, were labelled “Gap” and excluded from further analysis. Next, the conservation of the column across the whole alignment was calculated. Detecting conserved residues was based on information theory. Following Capra and Singh [61], the Shannon entropy of a column (i.e., a position in the multiple protein sequence alignment) was calculated. For amino acids in a column, entropy can take values between zero (all sequences are the same amino acid) and log_2_(20) (each amino acid is equally likely). If entropy was lower than 2/3 of a bit, the column was counted as conserved.

If the Shannon entropy of the column for the whole alignment was <2/3 of a bit, the column was labelled as “All proteins conserved”. Columns not labelled as “Gap” or “All proteins conserved” were then analysed by protein, e.g., the Shannon entropy was calculated for the column just in the ABCG1 sequences, or ABCG4 sequences. If it was not conserved (i.e., if the Shannon entropy within any of the proteins was <2/3) the position in the alignment was labelled “Not Conserved”. If a column was conserved in one or more proteins, the most common residue found in each protein was recorded. Each of these columns was recorded as a list of pairs of conserved residues and the proteins matching that residue at that column. For example, column 1011 in the alignment corresponds to the well-studied residue 482 in ABCG2. This is conserved in all ABCGs, but differently—in ABCG1 and ABCG4, it is glutamine; in ABCG5, it is serine; and in ABCG8, it is histidine—so the record for that column is:

(1011, [(‘R’, [ABCG2]), (‘S’, [‘ABCG5’]), (‘Q’, [‘ABCG1’, ‘ABCG4’]), (‘H’, [‘ABCG8’])]).

To display a summary of sequences, logos were constructed using LogoMaker [62]. The positions in a protein corresponding to columns of interest were displayed on the structure of ABCG2 PDBID: 6VXF [15] using ChimeraX [63].

### 4.4. Binding Pockets

Residues corresponding to the binding pocket of ABCG2 for imatinib, mitoxantrone and SN38 were identified by taking residues with any part 5 Å from the substrate in structures 6VXH, 6VXI, and 6VXJ, respectively. Residues contributing to potential binding pockets for cholesterol in ABCG5/G8 were taken from Lee et al. 2016 [9]. The columns corresponding to these residues were identified and compared.

### 4.5. Statistics

To estimate the threshold for significance for the number of columns with a given conservation, the probability of a column being conserved in a particular pattern was modelled as a Poisson distribution with λ of 595/202 (non-gap columns/possible conservation patterns). To find a threshold for significance for the 202 possible conservation patterns, an initial α = 0.1 was divided by 202. The cumulative probability of a conservation pattern occurring n times exceeds 1-(0.1/202) at 10 columns, so any conservation pattern with more than 10 columns was treated as significant.

The expected values for the frequency of each conservation pattern were based on the frequency of conserved residues for non-gap positions for each protein. First, assuming conservation between proteins is independent, the probability of any set of proteins being conserved was estimated as the product of probabilities of conservation for the proteins conserved multiplied by the product of the probability of each protein not conserved not being conserved.

For each of these sets, the possible conservation patterns were generated by finding all possible partitions of the set. The relative probabilities of each of these partitions was calculated assuming the residues were conserved for each column independently, so the probability of any two proteins conserving the same residue was 0.05. For each partition, the probability of a column being conserved that way, given that set of proteins is conserved, is then $\frac{19!}{\left(20-m\right)!20^{n-1}}$, where n is the size of the set and m is the number of parts. To obtain estimates for the expected value for each conservation pattern, these values were then multiplied by the probability of that set of proteins being conserved, then multiplied by the number of non-gap positions.

## Figures and Tables

**Figure 1 ijms-22-03012-f001:**
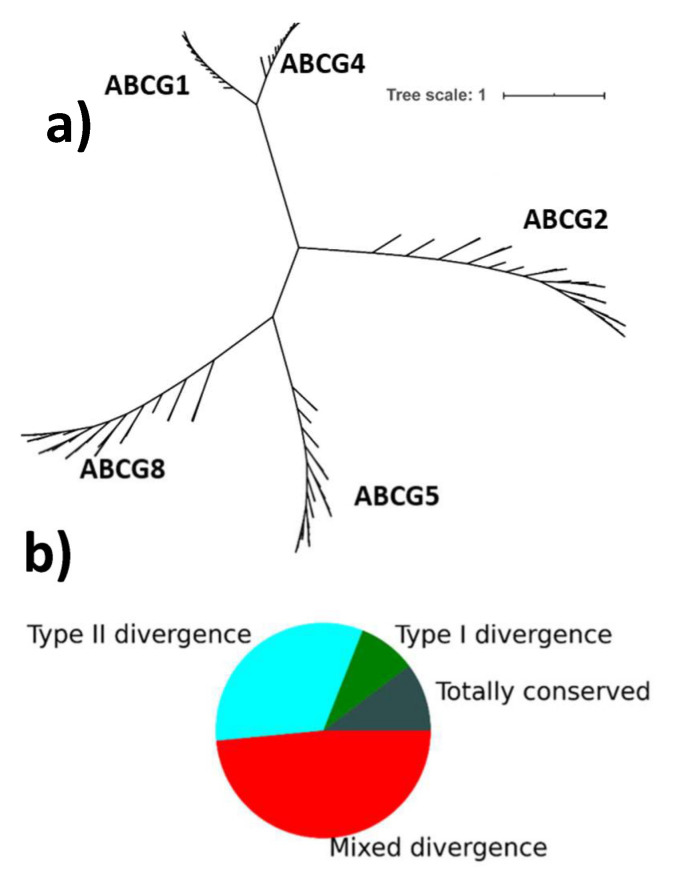
(**a**) Phylogenetic tree of mammalian ABC subfamily G proteins. Tree based on 174 protein sequences, aligned with multiple alignment fast Fourier transform (MAFFT). Names of taxa have been removed for clarity. (**b**) Pie chart showing proportions of conservation and divergence. In the 594 columns showing conservation in at least one protein in the G subfamily, 61 are totally conserved (grey); 52 show simple type I divergence (where one set has conservation, and the others do not) (green); 193 show type II divergence (where each set is conserved, but with a different residue) (cyan); and the remaining 288 have some mixture of divergence (e.g., column 891 is a conserved cysteine in ABCG2, and a conserved leucine in ABCG1 and ABCG4, but is not conserved in other groups. Thus it has neither purely type I nor type II divergence) (red).

**Figure 2 ijms-22-03012-f002:**
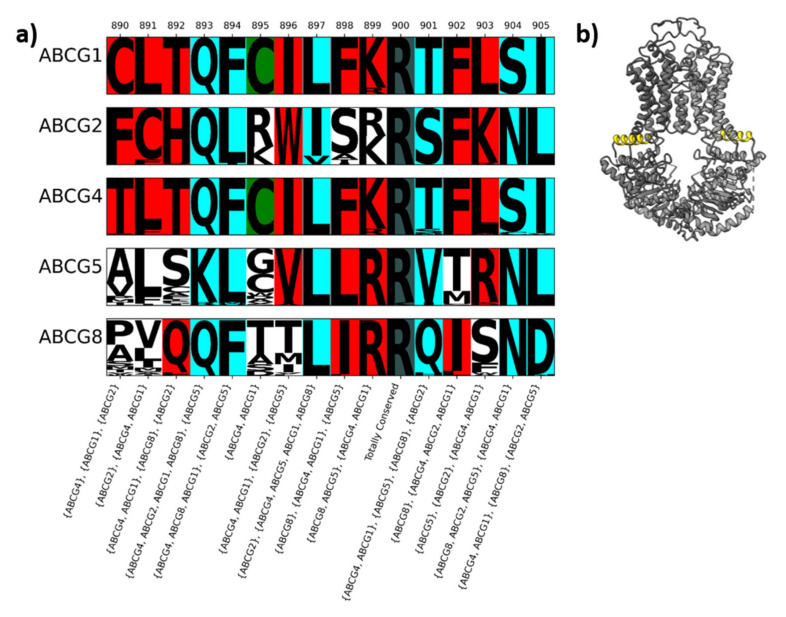
Conservation in the alignment of ABCG protein sequences. (**a**) Sequence logo in which sequences have been divided by the protein they represent. Font size corresponds to the fraction of sequences with that residue in that column. Conserved positions have coloured backgrounds so that totally conserved columns are grey, columns with type I divergence are green, columns with type II divergence are aqua, and columns with mixed divergence are red. Conservation patterns as described in the text are shown at the bottom. (**b**) Structure of ABCG2 (PDBID: 6vxf) highlighting the area represented in the logo. This corresponds to the elbow helix in ABCG2.

**Figure 3 ijms-22-03012-f003:**
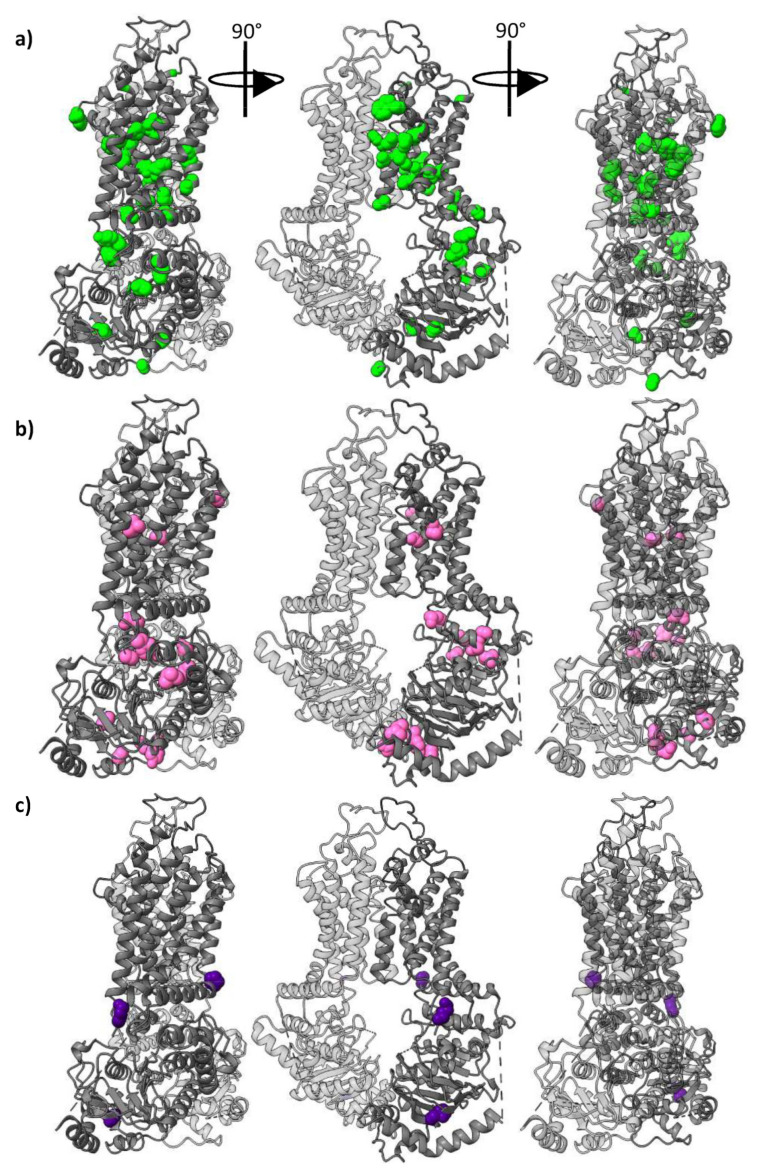
Functionally divergent residues shown on ABCG2 (PDBID: 6vxf) shown as coloured spheres on three views of the structure. (**a**) Residues conserved in the pattern (ABCG1, ABCG4), (ABCG2) (ABCG5), (ABCG8) as green spheres. (**b**) Residues conserved in the pattern (ABCG1, ABCG4), (ABCG2), (ABCG5, ABCG8) as pink spheres. (**c**) Residues conserved differently in each protein, i.e., with the conservation pattern (ABCG1), (ABCG4), (ABCG2), (ABCG5), (ABCG8) as purple spheres.

**Figure 4 ijms-22-03012-f004:**
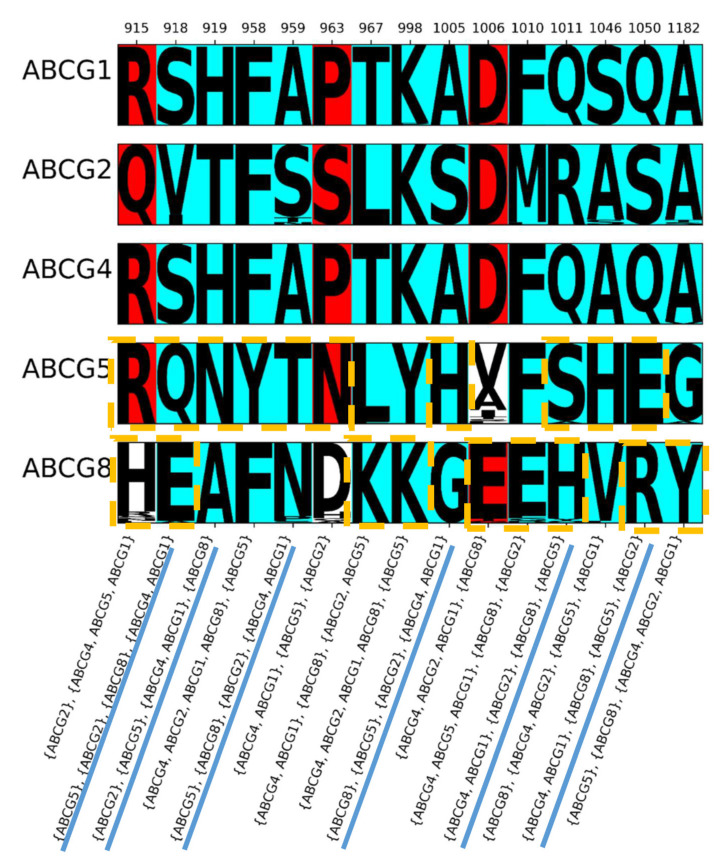
Conservation patterns in columns corresponding to the polar relay in ABCG5/G8. Columns coloured as in Figure 2. Orange dashed boxes indicate the polar relay for ABCG5 and ABCG8. Columns with the conservation pattern (ABCG1, ABCG4), (ABCG2), (ABCG5), (ABCG8) have that pattern underlined in blue at the bottom.

**Figure 5 ijms-22-03012-f005:**
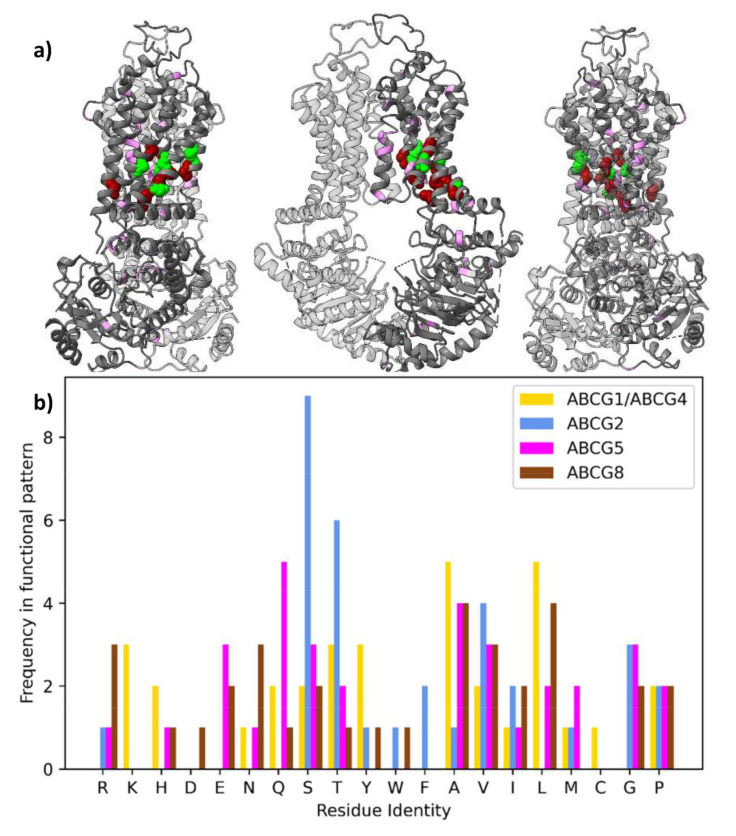
Comparison of the polar relay with functionally divergent residues. (**a**) Distribution on structure of ABCG2. Residues found in the polar relay are shown as spheres. Those with the conservation pattern (ABCG1, ABCG4), (ABCG2), (ABCG5), (ABCG8) are coloured green. Others are coloured red. Residues outside the polar relay with the conservation pattern above are coloured violet within the cartoon representation. (**b**) Identity of residues with conservation pattern (ABCG1, ABCG4), (ABCG2), (ABCG5), (ABCG8). Bars are coloured by protein, and their height represents the number of that residue found in the 33 positions with the above conservation pattern for that group. For each residue, bars are in the order ABCG1 and ABCG4; ABCG2; ABCG5; and ABCG8.

## Data Availability

Python code used in this article is available at https://github.com/kuraisle/ABCG_Family_Analysis (accessed on 15 March 2021), which also includes the sequence alignment used and instructions on using the code to explore it.

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
