# Peer review of "Analysis of Sequence Divergence in Mammalian ABCGs Predicts a Structural Network of Residues That Underlies Functional Divergence"

_ijms, 2021, doi:10.3390/ijms22063012_

Round 1
Reviewer 1 Report
The study is well done, the material is large enough and the methods look reliable. However the study is based on extensive and very recent literature, gives some new information and this warrants its publication.
Author Response
Reviewer 1 had no corrections for us to make.
Reviewer 2 Report
The authors performed in silico analyses of sequence divergence in mammalian ABCG sub-family members to explore if differences in conservation have a correlation with functional variation. They identified important conserved residues in ABCG subfamily members, whose conservation suggests an important role in substrate specificity. When the authors mapped sets of conserved patterns including the most common pattern [(ABCG1, ABCG4), (ABCG2), (ABCG5), (ABCG8)] onto the resolved ABCG2 structure, it revealed a "corkscrew" pattern. Importantly, this pattern in ABCG2 is ideally placed to form a network to coordinate conformational changes throughout the protein and provide additional conformational flexibility and confer broad substrate specificity.
Specific comments:
- In line 324, the authors mentioned "though no experimental work has yet been carried out to deliberately explore the functional effects of mutations of the residues of the corkscrew, some of these have been mutated as part of other studies, or observed as naturally occurring single nucleotide polymorphisms.” However, the prediction of functional effect based on in silico analyses might not be supported by the biochemical studies. The authors should address these issues:
- If we consider the conserved N418 residue (based on the authors' analysis) in ABCGs, published work suggests that mutation of N418 to a similar residue Q does not change the expression or transport function of ABCG2 (Ref: Diop and Hrycyna, Biochemistry 2005). However, mutation of N418 to alanine could provide important information regarding the functional role of this conserved residue in ABCG2.
- The authors mentioned in line 340 "mutation of P480 to alanine, despite being a mutation to a residue conserved in all sequences used in this analysis, and with dramatic chemical differences, does not affect transport function of ABCG2". This supports the idea that conservation of residue(s) doesn't always indicate functional relevance.
- As the conservational divergence contributes to the functional difference in ABCGs, can the substrate specificity be eliminated? Specifically, will a particular ABCG transporter have substrate specificity like other family members if the conservation is maintained through mutagenesis study? The atomic structures of ABCG2 with the bound substrates imatinib, mitoxantrone, and SN38 ( PDB ID 6VXH, 6VXI, and 6VXJ) are available. First, it would be good to include a figure showing the residues conserved in the ABCG2 substrate-binding pocket that interact with three substrates (imatinib, mitoxantrone and SN38) in the left panel and the substrate-binding pocket of ABCG5/8 with residues interacting with sterol substrate. A discussion about the extent of the difference in the residues of the binding pockets of ABCG2 and ABCG5/8 would be very helpful to the reader.
- Some minor editing is required in the manuscript:
- In line 173, there is an additional spacing before the sentence "The approach…..within the alignment".
- In line 198, it seems "this" word is inappropriate in this sentence context.
- In line 326, there is a typo for the word "corkscrewa" though it would be "corkscrew."
- Check whether two different text fonts and sizes are used for reference # within the main manuscript text.
Author Response
Responses to the reviewer's comments are in bold
- In line 324, the authors mentioned "though no experimental work has yet been carried out to deliberately explore the functional effects of mutations of the residues of the corkscrew, some of these have been mutated as part of other studies, or observed as naturally occurring single nucleotide polymorphisms.” However, the prediction of functional effect based on in silico analyses might not be supported by the biochemical studies. The authors should address these issues:
- If we consider the conserved N418 residue (based on the authors' analysis) in ABCGs, published work suggests that mutation of N418 to a similar residue Q does not change the expression or transport function of ABCG2 (Ref: Diop and Hrycyna, Biochemistry 2005). However, mutation of N418 to alanine could provide important information regarding the functional role of this conserved residue in ABCG2.
N418 in ABCG2 is not considered in this paper. When examining this column (935 in the alignment), it is not particularly well conserved, except in ABCG1. The data from Diop and Hrycyna, then, is in line with expectations from the sequence analysis.
- The authors mentioned in line 340 "mutation of P480 to alanine, despite being a mutation to a residue conserved in all sequences used in this analysis, and with dramatic chemical differences, does not affect transport function of ABCG2". This supports the idea that conservation of residue(s) doesn't always indicate functional relevance.
We have added the sentence "This provides a cautionary example that care must be taken when interpreting these results." to make it clear that our results are not unequivocal.
- As the conservational divergence contributes to the functional difference in ABCGs, can the substrate specificity be eliminated? Specifically, will a particular ABCG transporter have substrate specificity like other family members if the conservation is maintained through mutagenesis study? The atomic structures of ABCG2 with the bound substrates imatinib, mitoxantrone, and SN38 ( PDB ID 6VXH, 6VXI, and 6VXJ) are available. First, it would be good to include a figure showing the residues conserved in the ABCG2 substrate-binding pocket that interact with three substrates (imatinib, mitoxantrone and SN38) in the left panel and the substrate-binding pocket of ABCG5/8 with residues interacting with sterol substrate. A discussion about the extent of the difference in the residues of the binding pockets of ABCG2 and ABCG5/8 would be very helpful to the reader.
The idea of manipulating substrate specificity using the implications of this paper as a guide for mutagenesis is an interesting one, and one we have hinted towards with the final sentence of the main manuscript. We're glad this reviewer agrees.
Regarding the binding pockets of ABCG2 and ABCG5/G8, new sections have been added to the results (lines 268-276), and the methods (lines 422-427); and a new supplementary figure (SF4) and table (ST6) have been added.
- Some minor editing is required in the manuscript:
- In line 173, there is an additional spacing before the sentence "The approach…..within the alignment". This has been removed
- In line 198, it seems "this" word is inappropriate in this sentence context. "this forms" has been changed to "these residues form"
- In line 326, there is a typo for the word "corkscrewa" though it would be "corkscrew." This has been corrected
- Check whether two different text fonts and sizes are used for reference # within the main manuscript text. A different font had been added by the reference manager. The numbers' fonts have been changed in line with the manuscript
Reviewer 3 Report
The authors have selected a representative set of ABCG sequences and based on sequence alignment identified different conservation patterns among the residues. The study appears robust and is well explained and the manuscript is well written. I have only a few small comments:
In figure 5 a) it is not very easy to see difference between green spheres and green residues. I suggest to use third color to make the difference more clear. The authors state that there is six charged residues in the polar relay of ABCG5, but based on 5 b) isn’t it just five charged residues?
On line 241, I was wondering why tyrosine was not included with serine and threonine?
The summary of mutations in the supplementary was a little unclear to me. Was it supposed to include references of all studies where these mutations have been tested? This would indeed be a very useful collection. Or where just a few references selected (on what basis)? There were several residues now without references, even though their mutation and tested effect in vitro has been reported.
From the text it is unclear to me if the polar relay amino acids align in the ABCGs. Now they were marked for ABCG5/8, but as they vary among these two sequences, it is unclear which are supposed to belong to the polar relay in the other proteins. Are the columns included in figure 4 representing also the polar relay for the other ABCGs or only for ABCG5/8? Could the figure be altered to include marking of the polar relay residues of the other proteins? The authors suggest that the polar relay impacts the allosteric recognition of substrates, but previously also speculated that the polar relay might not be important for drug interaction, but for structural rigidity and stability (Khunweeraphong et al 2017). Can the authors comment on this?
Just out of curiosity, would a structural alignment (between ABCG2, ABCG5 and ABCG8) differ greatly from the sequence alignment?
Reference:
Khunweeraphong, N., Stockner, T. & Kuchler, K. The structure of the human ABC transporter ABCG2 reveals a novel mechanism for drug extrusion. Sci Rep 7, 13767 (2017).
Author Response
- In figure 5 a) it is not very easy to see difference between green spheres and green residues. I suggest to use third color to make the difference more clear. The authors state that there is six charged residues in the polar relay of ABCG5, but based on 5 b) isn’t it just five charged residues?
In Figure 5a), the residues of the corkscrew outside the polar relay have been coloured violet
The reviewer is correct about the discrepancy between the text and the figure here. The figure is correct, and line 223 has been edited.
- On line 241, I was wondering why tyrosine was not included with serine and threonine?
Tyrosine was not included because of its aromaticity, and how this precludes it from providing the same flexibility as serine and threonine. We appreciate that the text does not make this clear and have edited the text on line 229 to make this clearer.
- The summary of mutations in the supplementary was a little unclear to me. Was it supposed to include references of all studies where these mutations have been tested? This would indeed be a very useful collection. Or where just a few references selected (on what basis)? There were several residues now without references, even though their mutation and tested effect in vitro has been reported.
The summary in the supplements was intended to be reasonably comprehensive, without being unwieldy. Where a small number of references exist for a mutant, we have endeavoured to be complete. Where there are a large number of references (for example for R482 in ABCG2) that reach similar conclusions, the list is truncated. If some mutations have been missed, it is despite a manual literature search.
- From the text it is unclear to me if the polar relay amino acids align in the ABCGs. Now they were marked for ABCG5/8, but as they vary among these two sequences, it is unclear which are supposed to belong to the polar relay in the other proteins. Are the columns included in figure 4 representing also the polar relay for the other ABCGs or only for ABCG5/8? Could the figure be altered to include marking of the polar relay residues of the other proteins? The authors suggest that the polar relay impacts the allosteric recognition of substrates, but previously also speculated that the polar relay might not be important for drug interaction, but for structural rigidity and stability (Khunweeraphong et al 2017). Can the authors comment on this?
The polar relay in the core of the TMDs is present in all ABCG transporters. The polar relay residues are explicitly identified in ABCG5 and ABCG8. These do not perfectly match between the two proteins, so the union of the two has been used to identify the polar relay of the whole family. The number of charged residues varies in the ladder of the polar relay, and therefore also charges and polarity, but the sidechains of the polar relay occupy the same region within the TMDs of ABCG transporters. A key feature of the polar relay is its structure, in that it is found in the core of the transmembrane part of a protein. Only a few residues oriented towards the cytosol or the TMD-TMD interface are accessible, most prominently residue Q398 and E446 (in ABCG2). From the energetic point of view, an important difference between charge-charge interactions and hydrophobic interactions is that charge-charge interactions are much less sensitive to small structural changes than hydrophobic interactions. As a consequence, the polar interactions provide stability, because the forces between charges are strong, but also allow for some flexibility as they vary less with distance. The polar relay in ABCG2 discussed in Khunweeraphong et al. is based on ABCG5/G8, as in this manuscript. The current manuscript represents an evolution of the role of the polar relay which has been has identified structurally, and is now shown to have a hallmark conservation pattern that aligns to functional divergence. True identification of its role needs to come from future site directed mutagensis studies which this paper hopes to incite.
- Just out of curiosity, would a structural alignment (between ABCG2, ABCG5 and ABCG8) differ greatly from the sequence alignment?
We tried this, using POSA, aligning the structures as available in the PDB. There were some minor differences, but most of the alignment was the same for all 3 chains, and where there were differences, they were small.
Reviewer 4 Report
The sequence analysis presented by the authors is interesting and useful for planning future experiments to test the relevance of the identified residues. However, the current version of the manuscript needs urgent reorganization, specially concerning the figures and their citation in the text. The disorganization makes it difficult to follow the idea of the manuscript in a straight-forward way. For example:
-In the text there is no reference to figure 3c or figure 5a. They are never mentioned or discussed in the text. Supplementary figure 3 is not referred to in the main manuscript either.
-Figure 3a is referred to in page 5 line 197, but figure 3b is mentioned in page 10, line 286, after the authors had already described figures 4 and 5.
-A similar disorganization is found for the supplementary information: currently the figures are shown first, followed by the tables. But some tables are referred to in the text before some of the figures, so the reader needs to jump from one place to the other several times. It is highly recommended to present the supplementary figures/tables in the order they are being mentioned in the manuscript.
-Minor: in page 10, the first word in line 326 is “screwa”. Should it be “screw”?
Additional comments:
-From the figures, there seems to be some overlap between the corkscrew and the polar relay. Are there any common residues between the “corkscrew” pattern and the polar relay? It would be useful to present in the supplementary information a list (or like figure 4) of the amino acids forming the “corkscrew” and the other two patterns shown in figure 3.
-The authors suggest that the larger proportion of S and T in the polar relay of ABCG2 could “form intra-helical hydrogen bonds, which can bend the helix in certain conformations, 248 lending ABCG2 unusual flexibility in this region”. Can the authors specify the identity of those residues and show them in a figure so the readers can observe what region(s) would be expected to undergo such suggested intrahelical bonds and bending?
-At the end of the discussion the authors write: “Particularly, grouping 351 ABCG1 and ABCG4 together identifies a pattern which is also important to a previously 352 identified structural feature and suggests a unifying hypothesis for substrate specificity 353 in the subfamily”. This is essentially their conclusion, so, can the authors be more specific about what was the identified structural feature and what is that unifying hypothesis they are referring to?
Author Response
- In the text there is no reference to figure 3c or figure 5a. They are never mentioned or discussed in the text. Supplementary figure 3 is not referred to in the main manuscript either.
Figure 3c is now referred to in line 191. Figure 5a is referred to in line 212. Supplementary figure 3 is now supplementary figure 5, mentioned in line 282.
- Figure 3a is referred to in page 5 line 197, but figure 3b is mentioned in page 10, line 286, after the authors had already described figures 4 and 5.
All panels in figure 3 are now mentioned in lines 183-191.
- A similar disorganization is found for the supplementary information: currently the figures are shown first, followed by the tables. But some tables are referred to in the text before some of the figures, so the reader needs to jump from one place to the other several times. It is highly recommended to present the supplementary figures/tables in the order they are being mentioned in the manuscript.
These have been reorganised to reflect the order in the manuscript.
- Minor: in page 10, the first word in line 326 is “screwa”. Should it be “screw”?
This has been corrected.
Additional comments: - From the figures, there seems to be some overlap between the corkscrew and the polar relay. Are there any common residues between the “corkscrew” pattern and the polar relay? It would be useful to present in the supplementary information a list (or like figure 4) of the amino acids forming the “corkscrew” and the other two patterns shown in figure 3.
Figures 4 and 5a, and lines 210-214 are explicit in identifying the significant overlap of the corkscrew network and the polar relay. To save the supplementary information from being an endless table, the available code, including an interactive version, can be used to explore any conservation pattern. Lines 465 and 466 have been edited to make it clear that this is available.
- The authors suggest that the larger proportion of S and T in the polar relay of ABCG2 could “form intra-helical hydrogen bonds, which can bend the helix in certain conformations, 248 lending ABCG2 unusual flexibility in this region”. Can the authors specify the identity of those residues and show them in a figure so the readers can observe what region(s) would be expected to undergo such suggested intrahelical bonds and bending?
Supplementary figure 3 now shows the serines and threonines of the corkscrew in the TMD of ABCG2.
- At the end of the discussion the authors write: “Particularly, grouping 351 ABCG1 and ABCG4 together identifies a pattern which is also important to a previously 352 identified structural feature and suggests a unifying hypothesis for substrate specificity 353 in the subfamily”. This is essentially their conclusion, so, can the authors be more specific about what was the identified structural feature and what is that unifying hypothesis they are referring to?
We have clarified this to reinforce the concept that by treating ABCG1 and ABCG4 together in sequence analyses, rather than separately, allows the identification of a signature helical screw of residues that are proposed to define substrate specificity in ABCG family transporters in lines 351-354.